# Otsu Multi-Threshold Image Segmentation Based on Adaptive Double-Mutation Differential Evolution

**DOI:** 10.3390/biomimetics8050418

**Published:** 2023-09-08

**Authors:** Yanmin Guo, Yu Wang, Kai Meng, Zongna Zhu

**Affiliations:** 1Shandong Research Institute of Industrial Technology, Jinan 250100, China; guoyanmin@sriit.cn (Y.G.); mengkai@sriit.cn (K.M.); 2School of Computer Science and Technology, Shandong University of Finance and Economics, Jinan 250014, China; 15665756530@163.com

**Keywords:** differential evolution, image segmentation, Otsu, threshold

## Abstract

A quick and effective way of segmenting images is the Otsu threshold method. However, the complexity of time grows exponentially as the number of thresolds rises. The aim of this study is to address the issues with the standard threshold image segmentation method’s low segmentation effect and high time complexity. The two mutations differential evolution based on adaptive control parameters is presented, and the twofold mutation approach and adaptive control parameter search mechanism are used. Superior double-mutation differential evolution views Otsu threshold picture segmentation as an optimization issue, uses the maximum interclass variance technique as the objective function, determines the ideal threshold, and then implements multi-threshold image segmentation. The experimental findings demonstrate the robustness of the enhanced double-mutation differential evolution with adaptive control parameters. Compared to other benchmark algorithms, our algorithm excels in both image segmentation accuracy and time complexity, offering superior performance.

## 1. Introduction

The technique of images segmentation involves breaking an image up into a number of distinct, non-overlapping parts and extracting the desired, human-interest-centered regions. From image processing to image analysis, it is a committed step that makes it easier for following computer vision, etc. There are other widely used image segmentation methods in addition to the Otsu threshold image segmentation method [1,2,3] in the field. The maximum entropy method, for instance [4]. A well-known edge detection strategy for extracting edge information from images is the Prewitt image segmentation technique [5]. This method identifies the edges in the image by detecting gradient changes and is based on gradient calculation of the grayscale values of image pixels. The watershed image segmentation method [6] is a region-based image segmentation method. In this method, the image is considered as a terrain map, where the height represents the grayscale value of the image. Image segmentation is achieved by finding watersheds, which divide the image into different regions or sets of different regions. The advantage of this method is that it can preserve areas with clear edges and rich textures in the image. However, there is also a drawback of over segmentation for images with large uniform areas. The Canny image segmentation method [7] is a classic edge detection algorithm that can accurately detect edges in an image. This algorithm determines image edges by detecting gradient changes, and improves the accuracy and continuity of edges through non maximum suppression, dual threshold processing, and edge connectivity. The Sobel image segmentation method [8] is a commonly used edge detection algorithm that is used to detect edges in images. The Sobel operator is used to calculate the gradient value of image pixels to determine the position and direction of the image. The Robert image segmentation method [9] is a classic edge detection algorithm that applies the Robert operator to convolution the image and calculates the gradient values between adjacent pixels to determine the position and direction of the edges. An image segmentation technique based on the Laplacian Gaussian filter is the log image segmentation approach [10]. By applying Gaussian and Laplacian filtering procedures to the image, this technique finds edges and other features in the image. The threshold image segmentation approach is a common one in image processing technology because of its benefits of simplicity, effectiveness, and strong robustness. Single-threshold image segmentation and multi-threshold image segmentation are two common threshold segmentation techniques. The threshold is determined using the Otsu threshold segmentation technique. The time complexity grows exponentially as the number of thresholds increases, as does the quantity of calculation.

One might think of the threshold image segmentation approach as an optimization issue. Researchers are increasingly merging threshold image segmentation with intelligent optimization algorithms as swarm intelligence optimization techniques grow over time. For instance, based on the enhanced Firefly algorithm [11], better particle swarm optimization algorithm [12], improved genetic algorithm [13], improved whale optimization algorithm [14], and improved cuckoo algorithm [15]. You can use these clever optimization methods to identify appropriate optimization objective functions. The Otsu method and the maximum entropy method are common objective functions used in image segmentation.

Differential evolution [16] (DE) was proposed by Storn and Price in 1997. Because of its small control parameters and strong robustness, differential evolution has been widely concerned by researchers. The basic operations of differential evolution include initial population, mutation, crossover and selection. As long as the population iteration conditions are met, the population will iterate continuously to find the optimal solution. Therefore, differential evolution is also a parallel search algorithm. In recent years, with the in-depth innovation of research, more and more researchers found that control parameters and mutation strategies are the main factors that affect the performance of differential evolution. The key control parameters in differential evolution mainly include population size NP, mutation operator F and crossover factor CR. Based on this, researchers continue to study and find variants of differential evolution. In reference [17], AC Sanderson et al. proposed an improved differential evolution (JADE) using a new mutation strategy of selecting mutation vectors from an external archived population, and the control parameters F and CR in the population obey Cauchy distribution and Normal distribution, respectively. Ryoji Tanabe et al. [18] developed an improved differential evolution algorithm (SHADE) based on JADE, which guides the selection of control parameters in subsequent iterations through the number of successful individuals with historical mutations. The experimental results showed that SHADE outperforms previous state-of-the-art DE algorithms on a large number of benchmark problems. Wu Deng et al. [19] proposed a differential evolution with a new hybrid mutation factor and adaptive control parameters. Experimental results show that compared with other DE variants, the algorithm has better performance on the above test functions.

The differential evolution algorithm (DE) is a population intelligent optimization algorithm that searches for the optimal solution by simulating the process of natural evolution. With the continuous deepening of research on differential evolution algorithms, their application fields are gradually expanding. More and more optimization problems are combined with differential evolution algorithms to solve the optimal solution, such as scheduling problems [20], traveling salesman problems [21], and image processing [22]. In scheduling problems, differential evolution algorithms optimize scheduling strategies and resource allocation to maximize work efficiency or minimize work time. Differential evolution combined with scheduling problems can handle complex constraint problems and quickly converge to find the optimal solution. The traveling salesman problem is a classic combinatorial optimization problem that requires finding the shortest path so that the traveling salesman can visit several cities and ultimately return to the starting city. The differential evolution algorithm searches for the optimal solution by adjusting the arrangement order of cities, with the goal of minimizing travel and finding the optimal solution for the shortest path. In this problem, differential evolution algorithm can efficiently search for the global optimal solution. In image processing problems, differential evolution algorithms are used for image registration, image segmentation, feature extraction, and image enhancement.

Bird swarm behavior is simulated via particle swarm optimization [23]. In the process of segmenting a picture, each particle represents a potential segmentation scheme, and the best segmentation outcome is sought by continual iteration. With the help of pheromone propagation and reinforcement techniques, the Ant Colony algorithm [24] eventually achieves the best segmentation outcomes by simulating the behavior of ants as they look for food. The genetic algorithm [25] mimics the evolution process in nature by repeatedly iterating the population through selection, crossover, and mutation operations, maintaining its diversity and exploratory capacity through mutation, and finally discovering the best answer. Due to its few control parameters, robustness, and difficulty finding optimal solutions, the differential evolution method [26] performs well and offers advantages in picture segmentation.

The differential evolution variant tested against pertinent benchmark functions performed really well. Researchers are increasingly using differential evolution to enhance and optimize image processing outcomes. To determine the ideal threshold and enhance the performance of image segmentation, Sushi L. Kumar et al. integrated differential evolution with the Otsu threshold segmentation method in reference [26]. The experimental results demonstrate that the application of differential evolution to image segmentation can significantly increase the segmentation quality when compared to the segmentation results obtained using the conventional Otsu threshold segmentation method. However, the overall performance of differential evolution depends on the ability to balance global search and local search. The reason for its greater impact is the selection of control parameters. However, the above differential evolution proposed by Sushi L Kumar et al. is applied to threshold image segmentation, with high time complexity and poor robustness. Unable to meet the processing objectives of multi-threshold image segmentation. In reference [27], Helen Vicente Humann Ayala et al. proposed an improved differential evolution (BDE) based on the generation of beta distribution adaptive control parameters F and CR values. In each iteration, the adaptive control parameters generate random F and CR values, enhancing the random search ability of the algorithm and improving the performance of the algorithm. Experimental results show that the improved differential evolution (BDE) is more effective than the FODPSO algorithm. However, due to the fact that the BDE algorithm randomly generates the values of control parameters F and CR, it cannot be ensured that the F and CR values are suitable for the entire iterative process of the algorithm. Based on the above algorithm. In this paper, a differential evolution based on adaptive double mutation is proposed for image segmentation. The population is divided according to the fitness value, and the double-mutation strategy is used to maximally balance the global search ability and local search ability of the algorithm. In addition, the values of control parameters F and CR are adaptively adjusted according to different stages of population evolution. The improved differential evolution in this paper is applied to Otsu multi-threshold image segmentation, and the maximum inter class variance method is regarded as the objective function to find the optimal threshold. The experimental results show that the algorithm is more accurate than other benchmark algorithms in image segmentation, with low time complexity and strong robustness.

Here, is the rest of the essay: The related work, such as the traditional differential evolution and Thresholding picture segmentation, are introduced in the second section. The third section introduces the TRDE algorithm proposed in this article. The fourth section introduces the experimental results, the fifth section is the analysis and research discussion of the experimental results, the sixth section is the references and future work, and the third part describes the improved double-mutation adaptive differential evolution algorithm (TRDE).

## 2. Related Work

### 2.1. The Classical DE Algorithm

Differential evolution is a population intelligent optimization algorithm. DE algorithm includes the following four steps: population initialization, mutation operation, crossover operation and selection operation. The following is the basic process of the DE algorithm:

(1)Population initialization

Randomly generate the initial population, initialize and generate NP individual vectors in d-dimensional space, denoted as:(1)$$x_{i,g}=(x_{1,g},x_{2,g},\ldots ,x_{d,g})\;i=1,2,\ldots ,NP$$

Among them, NP is the population size, and g is the current evolutionary algebra. The NP individual vectors generated in d-dimensional space are randomly generated, and the specific formula is as follows:(2)$$x_{i,j}=x_{i,j}^{L}+rand(0,1)\times (x_{i,j}^{U}-x_{i,j}^{L})$$

In the above formula, $x_{i,j}$ represents the ith randomly generated individual, $x_{i,j}^{U},x_{i,j}^{L}$ represents the Upper and lower bounds of the ith individual vector, rand(0,1) represents a random number generated within the range of (0,1).

(2)Mutation

The DE algorithm achieves individual vector mutation through differential strategy, which is also the core idea of the DE algorithm. Randomly select several individual vectors from the population for differential operation to generate differential vectors. Common mutation strategies are as follows:

DE/rand/1:(3)$$v_{i,g}=x_{r_{1},g}+F\times (x_{r_{2},g}-x_{r_{3},g})$$DE/rand/2:(4)$$v_{i,g}=x_{r_{1},g}+F\times (x_{r_{2},g}-x_{r_{3},g})+F\times (x_{r_{4},g}-x_{r_{5},g})$$DE/best/1:(5)$$v_{i,g}=x_{best,g}+F\times (x_{r_{1},g}-x_{r_{2},g})$$DE/best/2:(6)$$v_{i,g}=x_{best,g}+F\times (x_{r_{1},g}-x_{r_{2},g})+F\times (x_{r_{3},g}-x_{r_{4},g})$$
wherein $x_{r_{1},g},x_{r_{2},g},x_{r_{3},g},x_{r_{4},g},x_{r_{5},g}(r_{1}\neq r_{2}\neq r_{3}\neq r_{4}\neq r_{5})$ represent five randomly selected individual vectors from the population that are different from each other, $v_{i,g}$ represents the variation vector of the ith individual in the generated generation, and F is the scaling factor.

(3)Crossover

Crossover operation is to cross the generated mutated individual vector with the parent individual vector to generate an experimental individual vector. The commonly used binomial cross operation in DE algorithm is as follows:(7)$$u_{i,j,g}=\left\{ \begin{array}{l} v_{i,j,g}\;if\;rand\leq CR\;or\;j=j_{rand}\\ x_{i,j,g}\;\;otherwise \end{array}\right.$$

In the above formula, CR is the crossover operator, $j_{rand}$ is a random integer selected from {1, 2, …, d} to ensure that the mutated individual has at least one individual component inherited to the next generation, avoiding the same vector as the parent individual.

(4)Selection

The selection operation in the DE algorithm adopts a greedy strategy, which selects the individual vector with the best fitness value as the evolutionary offspring vector through comparison. The specific formula is as follows:(8)$$x_{i,g+1}=\left\{ \begin{array}{l} u_{i,g},if\;f(u_{i,g})\geq f(x_{i,g})\\ x_{i,g},otherwise \end{array}\right.$$

### 2.2. The Otsu Threshold Segmentation Method

The maximum inter class variance method [28] was proposed by Japanese scholar Otsu in 1997. The basic principle of Otsu for image threshold segmentation is to divide the image into background and target categories based on the grayscale characteristics of the image, calculate the number of pixels in the image, and solve for the maximum variance between the image target and background to achieve image threshold segmentation.

For a pair digital image of M × N, where i is the grayscale value, L is the grayscale level, and the range of grayscale values is [0, L − 1]. $n_{i}$ represents the number of pixels with grayscale level i, and the probability of grayscale level i appearing in the image is $p_{i}$, then:(9)$$P_{i}=\frac{n_{i}}{N},\;\;I=0,1,2,\ldots ,L-1$$
(10)$$\sum_{i=0}^{L-1}P_{i}=1$$

Based on the grayscale characteristics of digital images, the pixel points in the image are divided into two categories: target and background using threshold s, Represented by $H_{0}$ and $H_{1}$. The grayscale values between [0, s] are classified as $H_{0}$, and the remaining grayscale values between [s + 1, L − 1] are classified as $H_{1}$. For the entire image, the average grayscale value is:(11)$$Q_{s}=\sum_{i=0}^{L-1}iP_{i}$$

The mean of $H_{0}$ and $H_{1}$ is:(12)$$q_{0}=\sum_{i=1}^{s}\frac{ip_{i}}{t_{0}}$$
(13)$$q_{1}=\sum_{i=s+1}^{L-1}\frac{ip_{i}}{t_{1}}$$

In the above formula:(14)$$t_{0}=\sum_{i=0}^{s}p_{i}$$
(15)$$t_{1}=\sum_{i=s+1}^{L-1}p_{i}=1-t_{0}$$

Based on the above formula, we can obtain:(16)$$q_{s}=t_{0}q_{0}+t_{1}q_{1}$$

The definition of inter class variance is as follows:(17)$$\begin{array}{ll} \sigma_{B}^{2} & =t_{0}{\left(q_{0}-q_{s}\right)}^{2}+t_{1}{\left(q_{1}-q_{s}\right)}^{2}\\ & =t_{0}q_{0}^{2}+t_{1}q_{1}^{2}-q_{s}^{2}\\ & =t_{0}q_{0}^{2}+t_{1}q_{1}^{2}-{\left(t_{0}q_{0}+t_{1}q_{1}\right)}^{2}\\ & =t_{0}t_{1}{\left(q_{0}-q_{1}\right)}^{2} \end{array}$$

Through the above formula, we can see that the s value corresponding to the maximum inter class variance is the optimal segmentation threshold.

### 2.3. Existing Problems in Research and Motivation for Improvement

The segmentation criterion for traditional Otsu is the largest variance between the target and background of the picture, which divides the target and backdrop of the image into two categories. However, the histogram of the segmented Grayscale typically does not exhibit a bimodal form, making it challenging to obtain precise image segmentation findings. To obtain the ideal threshold, many researchers treat threshold picture segmentation as an optimization problem and pair it with clever optimization methods. The control settings and mutation approach in the traditional DE algorithm have the most effects on the algorithm’s overall performance. In the DE algorithm, the important control parameters are the scaling factor F and the crossover factor CR. The scaling factor F controls the magnitude of differential evolution vector, which affects the convergence ability and Rate of convergence of the algorithm. The crossover factor CR affects the probability of generating a vector of experimental individuals in crossover operations to inherit genes from their parents or mutated individuals. The second part is the improvement of mutation strategy, balancing the global search ability and local search ability of the algorithm. To sum up, this paper proposes an adaptive double-mutation differential evolution algorithm (TRDE), which divides the population according to the fitness value of the population, adopts corresponding mutation strategies for different sub-populations, and adaptively adjusts the values of control parameters F and CR to adapt to different stages of population evolution.

## 3. The TRDE Algorithm

Researchers have enhanced and created variants of the DE algorithm based on it in order to significantly balance the global and local search capabilities. There are primarily two components in the DE algorithm that greatly influence how well it performs. The first step is to adjust the values of the parameters F and CR. The scaling factor F determines the size of the differential evolution vector and has an impact on the algorithm’s capacity for convergence and rate of convergence. The possibility of creating a vector of experimental individuals in crossover operations who would inherit genes from their parents or modified individuals is influenced by the crossover factor CR. a variety of adaptive parameter control DE algorithm versions, such as in reference [29]. A differential evolution algorithm based on dual mutation strategy is proposed for image segmentation. The experimental results show that this algorithm is competitive with other improved DE algorithms in terms of optimization accuracy and convergence speed. Vladimir Stanovov et al. proposed a method in reference [30] to adjust the values of control parameters F and CR based on fitness values, and increase the biased Lemmer mean and LBR to improve its performance. The experimental results show that the algorithm performance has been greatly improved. The second part is the improvement of mutation strategy, balancing the global search ability and local search ability of the algorithm. Regarding the improvement of mutation strategy, there are also many variants of DE algorithm. For example, in reference [31], Wan-li Xiang et al. proposed an enhanced DE algorithm (EDE), which uses a reverse learning strategy to increase the diversity of the initial population, improves the mutation strategy on the basis of JADE algorithm, and improves the overall performance of the algorithm. As mentioned in reference [32], Ali Wagdy Mohamed proposed a mutation strategy based on triangular mutation rules, which greatly balances the global detection ability and local search ability of the population.

### 3.1. Dual Mutation Strategy

The main component of the DE algorithm is the mutation strategy, which can make the population’s individual evolution vectors and orientations display diversity. Based on the fitness scores of various population vectors, this article separates the population into two subpopulations. Mutation strategy 1 is used for populations with high fitness values, while mutation strategy 2 is used for populations with low fitness values. In general, it is simpler to locate the ideal solution close to people with high fitness values or those with low principle fitness values. Consequently, the dual mutation technique can balance population diversity and local search capability, speeding convergence.

After initializing the population, calculate the population fitness value, and divide the population into two subpopulations based on the fitness value. Subpopulation 1 stores the vector of individuals with high fitness values, while Subpopulation 2 stores the vector of individuals with poor fitness values. Improved mutation strategy 1:(18)$$v_{i,g}=x_{i,g}+F_{i}\times (x_{dbest,g}-x_{r_{1},g})+F_{i}\times (x_{r_{2},g}-x_{r_{3},g})$$

$x_{dbest,g}$ is a random selection of one of the top d% individual vectors from Subpopulation1 as the basis vector. In this article, the selection of the d parameter adopts a linearly decreasing function. The specific changes in parameter d are as follows:(19)$$d_{i}=\frac{d_{\min}}{d_{\max}}(1-\frac{gen}{genertations})+(\frac{d_{\max}-d_{\min}}{d_{\max}+d_{\min}})$$

This article set $d_{\max}=0.8,d_{\min}=0.4$, gen is the current population evolution algebra, and generations is the total evolution algebra. The variation of d parameter shows a decreasing trend with the increase of iteration times. As the number of iterations in the population continues to increase, the fitness values of individual vectors in the population continuously concentrate in a certain local region. Therefore, in order to improve the local convergence ability of the population and find the optimal solution, in the later stage of iteration, the selection of the optimal d% individual vector gradually decreases, accelerating population convergence and enhancing convergence ability. Generally speaking, individuals with high fitness values or poor principle fitness values are more likely to find the optimal solution, while individuals with poor fitness values can enhance population diversity. Therefore, the mutation strategy 2 proposed in this article is as follows:(20)$$v_{i,g}=x_{i,g}+F_{i}\times (x_{r_{1},g}-x_{r_{3},g})+F_{i}\times (x_{r_{2},g}-x_{r_{4},g})$$

Based on the fitness value, randomly select four distinct integers $r_{1},r_{2},r_{3},r_{4}$ in Subpopulation2, and generate a difference vector for mutation operation to improve population diversity.

### 3.2. Parameter Adaptation

The fixed values of the control parameters F and CR in the traditional DE algorithm make it impossible for them to adapt to the algorithm’s needs at various phases of evolution. As a result, adaptive parameter control techniques are being used by more and more DE variations. In general, a DE algorithm should have strong global search capabilities, retain population variety as much as feasible in the early stages of population evolution, and look for the world’s best answer. In the later stage of population evolution, as the number of iterations keeps rising, it becomes increasingly important to have strong local convergence capacity and accelerate convergence.

The scaling factor F in the DE algorithm is closely related to the Rate of convergence, which controls the amplitude of the difference vector. In the early stages of population evolution, a larger value of the scaling factor F is beneficial for maintaining population diversity and ensuring that the algorithm searches throughout the entire solution space. As the number of iterations continues to increase, in the later stage of evolution, the optimal solution of the individual vector gradually concentrates in a certain local region. It is necessary to gradually reduce the value of F and strengthen the local convergence ability of the algorithm. The crossover factor CR affects the probability of experimental individuals generated by crossover operations inheriting genes from mutated individuals or parent individuals. Generally speaking, in the early stages of algorithm evolution, to ensure the algorithm’s global search ability, the CR value is small. In the later stage of algorithm evolution, in order to improve the local search ability of the algorithm with a larger CR value, combined with changes in the fitness value of individual vectors in the population, this paper proposes adaptive parameter control as follows:(21)$$F_{i}=F_{\min}-\ln(\frac{F_{\max}}{F_{\min}})\times {\left(\frac{gen}{generations}\right)}^{2}$$
where gen represents the evolutionary algebra of the current population, and generations represents the total number of iterations, $F_{\max}$ represents the maximum value for setting the scaling factor, $F_{\min}$ represents the minimum value for setting the scaling factor, This article set $F_{\max}=0.7,F_{\min}=0.2$.

The changes in the scaling factor F proposed above show a decreasing trend, which is consistent with the convergence characteristics of differential evolution.
(22)$$CR_{i}=(1-\frac{CR_{\min}}{CR_{\max}})\times {\left(\frac{gen}{generations}\right)}^{2}+\ln\frac{CR_{\max}}{CR_{\min}}\times (\frac{CR_{\min}}{CR_{\max}-CR_{\min}})$$
where gen represents the evolutionary algebra of the current population, and generations represents the total number of iterations, $CR_{\max}$ represents the maximum value for setting the crossover factor, and $CR_{\min}$ represents the minimum value for setting the crossover factor. In this article, set $CR_{\max}=0.9,CR_{\min}=0.1$. The change in CR in the above formula shows a monotonic increasing trend, which greatly balances the contradiction between population diversity and Rate of convergence. From the test results, it can be seen that the improved adaptive control parameter method has advantages over other benchmark algorithms.

### 3.3. Pseudo Code of TRDE

1:Initialize population *p* individuals and calculate their fitness values, NP = 30, gen = 1, generations = 100;2:for gen = 1 to generations do3:
for i = 1 to NP do4: 


 $F=F_{\min}-\ln(\frac{F_{\max}}{F_{\min}})\times {\left(\frac{gen}{generations}\right)}^{2}$
5: 


 $CR=(1-\frac{CR_{\min}}{CR_{\max}})\times {\left(\frac{gen}{generations}\right)}^{2}+\ln\frac{CR_{\max}}{CR_{\min}}\times (\frac{CR_{\min}}{CR_{\max}-CR_{\min}})$
6:

 Implement mutation in Equations (18) and (20);7:

 Implement crossover in Equation (7);8:
  Implement selection in Equation (8);9:
  end for10:
  gen = gen + 1;11:
end for

## 4. Experiment

### 4.1. Benchmarking Datasets and Benchmarking Algorithms

Better adaptive double mutation; in this work, the segmentation performance of the differential evolution (TRDE) method is evaluated using five thresholds between [2,6] and eight test images from the Berkeley database, along with the other five benchmark algorithms. The JADE method with archive operations, the SHADE algorithm, which is an improvement of JADE, the BDE algorithm, the HSDE algorithm with adaptive parameter control [33], and the SAF-DE algorithm [34] are some of the benchmark algorithms utilized in this article.

The population size NP in this article is universally fixed to 30 and the maximum number of iterations is set to 100 in order to guarantee the validity of the experimental results. The TRDE algorithm and five other benchmark algorithms are each run 30 times for the chosen 8 test images. Finally, the experimental impact of picture segmentation is assessed using evaluation indicators. Figure 1 illustrates the picture segmentation outcomes of the TRDE method for 8 photos and 5 thresholds within the range of [2,6]. Additionally, three comparing methods (BDE, JADE, and SAF-DE) were chosen at random for this study in order to display the segmented picture findings in Figure 2, Figure A1 and Figure A2.

This article not only compares the improved TRDE algorithm with five other benchmark algorithms applied to Otsu threshold segmentation, but also compares the TRDE algorithm with other image segmentation methods (Prewitt, Sobel, Canny, Robert) for image segmentation, and uses evaluation indicators for quantitative statistics.

### 4.2. Common Image Segmentation Quality Evaluation Standards

In this study, we used benchmark techniques for image segmentation as well as the improved adaptive double-mutation differential evolution. According to references [35,36], we discovered that the structural similarity (SSIM) and the peak signal-to-noise ratio (PSNR), as well as the standard deviation of fitness value and the running time of the algorithm, are the most frequently used image segmentation quality evaluation criteria.

(1)Structural Similarity (SSIM)

Digital images have certain similar features in space, and each pixel has correlation between them. Namely, structural similarity, by comparing structural information, further examining the distortion of the image. SSIM is a measure of image distortion based on human visual characteristics, defined as follows:(23)$$SSIM(x,y)=\frac{\left(2\mu_{x}\mu_{y}+C_{1})(2\sigma_{x,y}+C_{2}\right)}{\left(\mu_{x}^{2}+\mu_{y}^{2}+C_{1})(\sigma_{x}^{2}+\sigma_{y}^{2}+C_{2}\right)}$$

In the above formula, $\mu_{x}$ and $\mu_{y}$ represent the mean of x and y, respectively, while $\sigma_{x}$ and $\sigma_{y}$ represent the variance of x and y. $C_{1}$ and $C_{2}$ are two positive constants. The value range of SSIM is 0 to 1. In practical operation, it is necessary to first unify the image and also perform grayscale processing on the image. Usually, the larger the SSIM value, the higher the structural similarity between two images. The SSIM value of the same two images is equal to 1.

(2)Peak signal-to-noise ratio (PSNR)

The peak signal-to-noise ratio (PSNR) is an objective standard for evaluating images, typically used to measure the quality of compressed reconstructed images, measured in decibels (DB). Due to the fact that PSNR is based on error sensitive image quality evaluation and does not take into account the visual characteristics of the human eye, there may be inconsistencies between measurement results and subjective perception. The definition of PSNR is as follows:(24)$$PSNR=10\log_{10}\frac{255^{2}}{MSE}$$

MSE refers to the mean squared error of the original image and the segmented image. The definition of MSE is as follows
(25)$$MSE=\frac{\sum_{i=1}^{M}\sum_{j=1}^{N}{\left(x(i,j)-y(i,j)\right)}^{2}}{M\times N}$$

In the above formula, x(i, j) and y(i, j) represent the original image and the segmented image, respectively, while MSE represents the average of the sum of squares of the pixel values of the original image and the segmented image. Generally speaking, the higher the PSNR value, the higher the similarity between the original image and the segmented image, and the higher the segmentation quality of the resulting image.

The population size and number of iterations for the improved adaptive double-mutation differential evolution (TRDE) and other benchmark algorithms (JADE, SHADE, BDE, HSDE) in this study are set to the identical values. Run to get the segmented image and ideal threshold for each test image. To assess its segmentation effect, we employ the widely used evaluation criteria PSNR and SSIM for picture segmentation quality. Standard deviation and the algorithm’s average execution time should be used to assess the algorithm’s performance.

Figure 2 shows the segmentation results obtained by the TRDE image segmentation algorithm and four other image segmentation methods. It can be seen that the TRDE algorithm preserves as much structural information of the original image as possible by selecting the optimal threshold, resulting in better image segmentation results. The images segmented by other edge detection algorithms may have noise or rough edges, and the segmentation effect is shown in Figure 2.

## 5. Discussion and Analysis of Experimental Results

### Result Analysis

This study selected test images from the Berkeley segmentation dataset with each image size of 200,200, applied the TRDE algorithm and other benchmark algorithms to image segmentation, and evaluated the segmented images using evaluation indicators PSNR and SSIM. This study then conducted testing and analyzed experimental results. Table 1 displays the obtained PSNR data, while Table 2 displays the SSIM results.

These two evaluation indicators need to be considered together to evaluate the segmentation effect of an image. The larger the PSNR value and SSIM value, the better the segmentation effect of the image. The smaller the value, the worse the segmentation effect. Table 1 shows the peak signal-to-noise ratio (PSNR) calculated by the TRDE, JADE, SHADE, BDE, and HSDE algorithms for 8 test images at different thresholds. From Table 1, we can see that the segmentation performance of the TRDE algorithm is overall superior to other benchmark algorithms, with the PSNR values of the vast majority of segmented images ranking first. For Baboon images, when the thresholds are 3, 4, and 6, the PSNR values of image segmentation rank first. For camera images, only when the threshold is 5, the PSNR values of image segmentation rank second, and the rest of the PSNR values rank first. For pepper images, when the thresholds are 2, 3, and 4, the PSNR values of image segmentation rank first. For Saturn images, when the thresholds are 2, 3, 5, and 6, the PSNR values of image segmentation rank first. However, there are very few cases where the quality of image segmentation is poor. For example, when the pepper image threshold is 5, 6, the PSNR value of image segmentation ranks third and fourth. This is because the essence of differential evolution is a random algorithm, which may lead to poor image segmentation results.

Table 2 compares the segmented picture’s Structural Similarity Index (SSIM) to the effects of other benchmark algorithms JADE, SHADE, BDE, and HSDE on image segmentation. All SSIM scores for the Barbara image rank top for the 8 test images that were chosen. At thresholds of 3, 4, and 6, the TRDE algorithm performs the best image segmentation for the Lena image. According to the experimental findings, the TRDE algorithm successfully segments the great majority of images.

In comparison to other picture segmentation techniques, Table 3 displays the TRDE algorithm’s findings for the Structural Similarity Index (SSIM). The results of the TRDE algorithm applied to picture segmentation are shown in Table 4 along with comparisons to other image segmentation techniques. In contrast to standard SSIM, MS-SSIM evaluates the final assessment result by comparing the similarity of images at several scales and weighting the similarity at various scales. This paper presents an improved differential evolution technique for threshold image segmentation, which chooses an optimal threshold to preserve as much of the structural information of the original image as possible after segmentation. Due to the possible presence of noise or rough edges in the images segmented by edge detection algorithms, the structural similarity between the segmentation results and the original image may decrease. Therefore, the experimental results of the SSIM and MS-SSIM indicators for the segmented images of the TRDE algorithm in Table 3 and Table 4 are higher than those of the other four edge detection methods (Prewitt, Sobel, Canny, Robert).

The similarity index (UQI) of the TRDE algorithm used for picture segmentation is displayed in Table 5 along with comparisons to other image segmentation techniques. The UQI score ranges from 0 to 1, and a greater number denotes a higher degree of similarity and picture quality between the two photos. The UQI indicator thoroughly takes into account the mean, variance, and covariance of each image in order to assess how similar and high-quality each image is. Table 5 displays the experimental outcomes of the UQI picture quality evaluation index. The table shows that the TRDE algorithm consistently performs well when compared to other picture segmentation techniques. The Canny algorithm, for instance, places first and the TRDE method, second in photos such as Barbara and Lena. In cameraman images, the TRDE algorithm ranks first. Overall, TRDE has strong stability when applied to image segmentation algorithms, resulting in good segmentation image quality.

Taking into account the experimental results in Table 3, Table 4 and Table 5, the quantitative results show that the improved differential evolution algorithm (TRDE) applied to image segmentation has better image segmentation quality, strong algorithm robustness, and fast convergence speed compared to other image segmentation methods.

The enhanced adaptive double-mutation differential evolution (TRDE) and the other five benchmark algorithms (JADE, SHADE, BDE, SAF-DE, HSDE) are shown in Table 6 along with their average running times. The TRDE algorithm’s average running time is generally faster than other algorithms’ for photos of peppers and women. When the threshold is 2 or 3, the TRDE algorithm’s average running time for plane images is marginally greater than that of other techniques. Overall experimental findings demonstrate that this method’s rate of convergence is quicker than that of the other five benchmark algorithms (JADE, SHADE, BDE, SAF-DE, HSDE).

Table 1, Table 2, Table 6 and Table 7 illustrate the overall effectiveness of the enhanced adaptive double-mutation differential evolution (TRDE) in this study. The usual segmentation quality evaluation standards PSNR and SSIM are used to assess the segmented image. The improved adaptive double-mutation differential evolution has greater performance than previous benchmark algorithms, as can be observed from the experimental data listed in the table. Table 1 and Table 2 demonstrate that, when used for image segmentation, the method has good segmentation capabilities. Table 6 demonstrates that the enhanced differential evolution (TRDE) is more effective and has a faster Rate of convergence than previous algorithms. Table 7 demonstrates the TRDE algorithm’s strong resistance to the Otsu threshold image segmentation fitness function.

## 6. Conclusions

The adaptive double-mutation differential evolution method (TRDE) for picture segmentation is proposed in this research and employs Otsu as the goal function. In addition, compare this approach to five other benchmark algorithms (JADE, SHADE, BDE, SAF-DE, and HSDE), compute PSNR and SSIM values for the segmented images, determine the average algorithm running time, and thoroughly evaluate the technique’s performance. According to the experimental findings, the TRDE algorithm performs the best overall inside the [2,6] threshold range. Finally, the robustness of the method to the fitness function of Otsu threshold image segmentation was tested using the standard deviation of fitness values.

In future work, we will carry out studies from two aspects, improving different mutation strategies and optimizing the impact of adaptive control parameters on threshold selection for differential evolution. On the other hand, different threshold segmentation techniques are used to determine the optimal threshold.

## Figures and Tables

**Figure 1 biomimetics-08-00418-f001:**
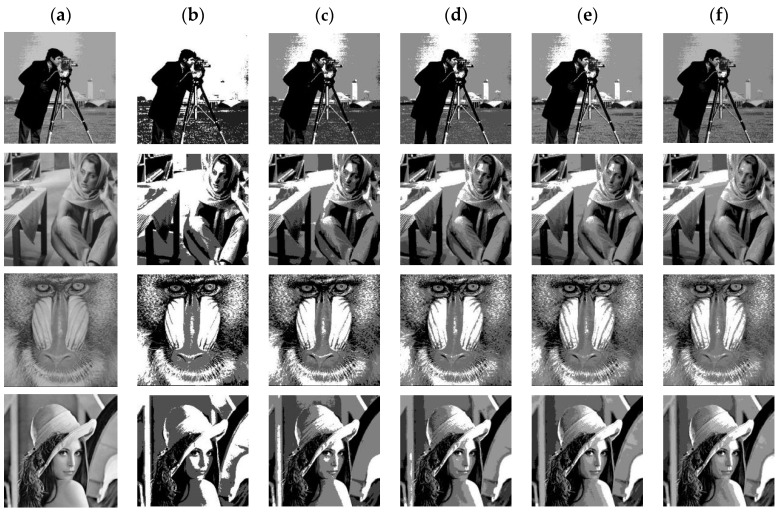
(**a**) Represents the original image, (**b**) segmented image with 2 thresholds, (**c**) segmented image with 3 thresholds, (**d**) segmented image with 4 thresholds, (**e**) segmented image with 5 thresholds, (**f**) segmented image with 6 thresholds (TRDE).

**Figure 2 biomimetics-08-00418-f002:**
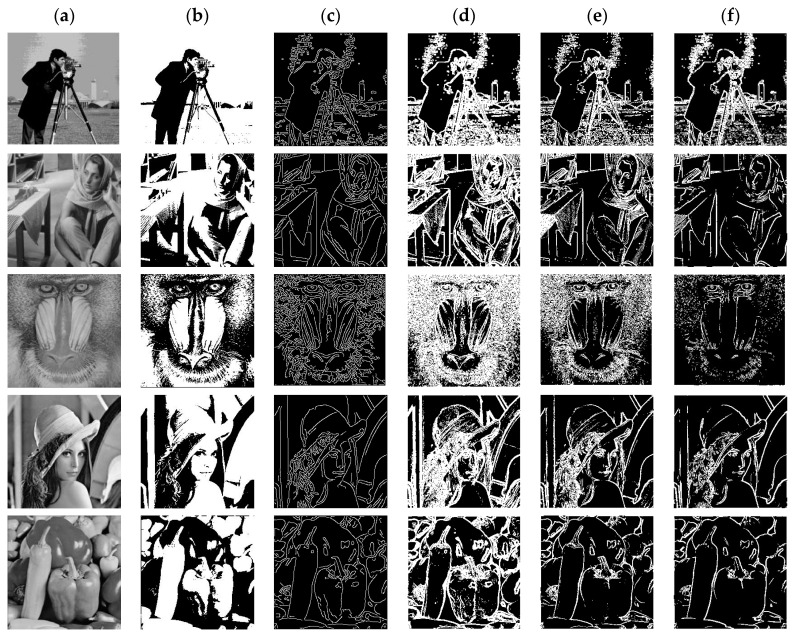
(**a**) Represents the original image and (**b**) the segmented image with TRDE, (**c**) the Canny image segmentation method, (**d**) the Sobel image segmentation method, (**e**) the Robert image segmentation method, and (**f**) the Prewitt image segmentation method.

**Table 1 biomimetics-08-00418-t001:** PSNR metrics for the JADE, SHADE, TRDE, SAF-DE, BDE and HSDE.

Images	K			Mean PSNR Value				
		JADE	SHADE	TRDE	SAF-DE	BDE	HSDE	Rank
baboon	2	16.7378	15.9039	16.4391	15.6738	15.5456	15.7254	2
	3	16.3789	16.0976	17.8936	16.0986	16.8789	16.7836	1
	4	19.2585	18.3252	19.3562	19.0358	19.2329	19.0231	1
	5	21.2548	20.1899	22.8248	21.9078	22.9806	21.4678	2
	6	22.3584	20.3145	23.0254	21.3023	22.1759	21.3985	1
barbara	2	18.7526	16.5962	18.3695	18.2036	18.2684	18.3206	2
	3	20.2562	19.3654	21.1485	20.3698	21.0369	21.0589	1
	4	22.2562	19.2585	22.5895	21.1485	21.6584	21.8548	1
	5	23.5412	20.6285	24.5621	22.6541	24.6254	22.1523	2
	6	23.0956	20.6598	24.3564	23.4589	24.2136	23.6984	1
cameraman	2	15.3405	14.4698	16.4223	15.4242	15.1344	15.1144	1
	3	18.6464	17.4566	18.7352	17.4547	17.4554	18.5468	1
	4	19.6876	18.4645	20.4687	20.4564	20.4578	19.4527	1
	5	21.2524	20.4546	22.4540	22.8676	22.4562	20.7462	2
	6	20.4766	20.7862	22.6786	21.4545	22.4564	21.4532	1
lena	2	19.8757	18.4666	20.4566	19.4646	19.4534	19.6783	1
	3	19.4587	18.4534	20.4782	19.7822	20.8786	19.4572	2
	4	20.8677	19.4568	20.8778	20.7864	21.4564	20.7862	2
	5	22.7862	21.7867	23.7676	22.2135	22.7824	21.4554	1
	6	22.7862	21.8672	23.7867	22.7862	23.4278	22.7240	1
peppers	2	20.2542	19.4534	20.5538	20.1230	20.2013	20.4878	1
	3	23.4564	22.4545	23.4776	23.1463	23.4564	23.4533	1
	4	24.4548	23.5821	25.4867	25.4534	23.4534	24.4532	1
	5	26.7863	24.4534	26.4564	27.4646	27.4564	25.1320	4
	6	24.4364	23.4534	26.1355	26.5368	26.4533	25.3451	3
plane	2	17.1355	17.1355	17.1515	17.1235	17.1351	17.2433	1
	3	18.3452	18.3456	18.3513	18.3153	17.3531	18.5678	2
	4	23.3153	19.1355	24.1353	23.3158	24.3153	21.4564	2
	5	23.3459	21.4648	25.3153	24.3531	24.3534	23.4556	1
	6	23.3453	23.3483	23.4155	23.3453	23.1354	23.6833	1
saturn	2	17.4324	17.4564	18.3455	17.3431	17.3154	17.4545	1
	3	19.3453	19.3453	20.4563	19.2112	19.2133	19.7896	1
	4	21.5645	21.3453	22.2543	22.3543	22.1546	22.3453	3
	5	22.4546	22.3122	23.4128	23.1234	23.3434	23.3456	1
	6	22.4545	22.5456	23.3432	22.9256	22.1531	22.4547	1
woman	2	15.3123	16.5467	16.4565	16.4532	16.2135	16.4534	2
	3	19.3434	18.4564	19.5369	19.4512	19.3155	19.4829	1
	4	21.3453	20.4565	21.3355	21.1233	21.3541	21.1442	3
	5	22.3468	21.3446	22.1238	21.3453	21.1243	21.7978	2
	6	21.8786	21.4678	22.3431	20.3453	21.2154	21.7828	1

**Table 2 biomimetics-08-00418-t002:** SSIM metrics for the JADE, SHADE, TRDE, SAF-DE, BDE and HSDE.

Images	K		Mean SSIM Value					
		JADE	SHADE	TRDE	SAF-DE	BDE	HSDE	Rank
baboon	2	0.64553	0.65347	0.68348	0.63453	0.63877	0.67865	1
	3	0.74534	0.75866	0.75877	0.71238	0.78767	0.78324	3
	4	0.83456	0.81234	0.83345	0.83455	0.81437	0.81537	2
	5	0.86455	0.88434	0.91433	0.90453	0.87373	0.82245	1
	6	0.94483	0.85665	0.94564	0.97684	0.94568	0.89786	2
barbara	2	0.38764	0.44678	0.45464	0.39448	0.38465	0.34646	1
	3	0.54645	0.44455	0.57876	0.57864	0.57846	0.56776	1
	4	0.70453	0.58886	0.71344	0.69874	0.71044	0.69854	1
	5	0.84453	0.69456	0.87213	0.82465	0.85345	0.72453	1
	6	0.80453	0.68764	0.89464	0.81345	0.88644	0.77646	1
cameraman	2	0.40456	0.45645	0.44545	0.43454	0.41343	0.3464	2
	3	0.56864	0.54615	0.59456	0.58456	0.54565	0.64535	2
	4	0.61534	0.65312	0.64631	0.68946	0.66464	0.70134	5
	5	0.72125	0.70456	0.74568	0.74561	0.74564	0.74562	1
	6	0.74565	0.74564	0.79563	0.79735	0.74564	0.77645	2
lena	2	0.64566	0.55445	0.65348	0.64556	0.64564	0.65434	2
	3	0.84561	0.84676	0.86462	0.74678	0.84564	0.84896	1
	4	0.86878	0.84345	0.87354	0.82345	0.86876	0.83455	1
	5	0.90432	0.89456	0.88473	0.89845	0.89854	0.88456	5
	6	0.88456	0.84562	0.90345	0.88456	0.86456	0.88455	1
peppers	2	0.49456	0.44564	0.55464	0.54564	0.54562	0.53245	1
	3	0.53455	0.54545	0.54567	0.58765	0.5345	0.53452	2
	4	0.53453	0.54534	0.54544	0.55433	0.53433	0.54524	2
	5	0.53153	0.55343	0.54354	0.54533	0.51234	0.57866	3
	6	0.56456	0.54534	0.56453	0.56467	0.56456	0.56465	5
plane	2	0.14564	0.18676	0.19564	0.15454	0.15464	0.14564	1
	3	0.28666	0.24568	0.25466	0.24564	0.25645	0.22343	3
	4	0.77456	0.42645	0.79465	0.74564	0.78678	0.47864	1
	5	0.78456	0.74654	0.88456	0.85664	0.75566	0.74424	1
	6	0.74568	0.74678	0.84568	0.78745	0.76456	0.74566	1
saturn	2	0.84566	0.84564	0.85475	0.85466	0.81564	0.85476	2
	3	0.84564	0.85464	0.88766	0.84564	0.84564	0.84564	1
	4	0.98764	0.84564	0.84564	0.86456	0.86456	0.84564	4
	5	0.95434	0.84543	0.95443	0.94564	0.84564	0.84534	1
	6	0.95343	0.95434	0.97163	0.93454	0.95464	0.94564	1
woman	2	0.36466	0.36866	0.36456	0.35646	0.36454	0.36456	3
	3	0.46546	0.45464	0.46556	0.45646	0.45466	0.46456	3
	4	0.45646	0.46456	0.48655	0.44568	0.44564	0.44566	1
	5	0.55641	0.55464	0.55466	0.46466	0.56456	0.56466	4
	6	0.56567	0.53245	0.74313	0.66466	0.67645	0.54566	1

**Table 3 biomimetics-08-00418-t003:** SSIM metrics for the TRDE, Canny, Prewitt, Robert and Sobel methods.

Images	Average SSIM Values of TRDE and Other Segmentation Methods				
	TRDE	Canny	Prewitt	Robert	Sobel
baboon	0.29329	0.00431	0.00075	0.00013	0.00363
barbara	0.33279	0.01241	0.01124	0.01231	0.00224
cameraman	0.61859	0.12219	0.11320	0.13265	0.10808
lena	0.42494	0.01050	0.00564	0.01130	0.00046
peppers	0.35451	0.01830	0.00462	0.00232	0.00124
plane	0.63998	0.00572	0.00525	0.00348	0.00839
saturn	0.73260	0.53086	0.50650	0.50648	0.49680
woman	0.39927	0.01021	0.00432	0.01436	0.00868

**Table 4 biomimetics-08-00418-t004:** MS-SSIM metrics for the TRDE, Canny, Prewitt, Robert and Sobel methods.

Images	Average MS-SSIM Values of TRDE and Other Segmentation Methods				
	TRDE	Canny	Prewitt	Robert	Sobel
baboon	0.29904	0.00252	0.00176	0.00263	0.00634
barbara	0.33919	0.01056	0.01054	0.00547	0.31222
cameraman	0.60684	0.12314	0.11637	0.13835	0.11504
lena	0.42384	0.00861	0.00479	0.00835	0.00339
peppers	0.35338	0.01537	0.00054	0.00264	0.00203
plane	0.63590	0.00322	0.00384	0.00548	0.01415
saturn	0.72052	0.54405	0.51520	0.51948	0.51394
woman	0.38647	0.00535	0.01222	0.02372	0.01807

**Table 5 biomimetics-08-00418-t005:** UQI metrics for the TRDE, Canny, Prewitt, Robert and Sobel methods.

Images	Average UQI Values of TRDE and Other Segmentation Methods				
	TRDE	Canny	Prewitt	Robert	Sobel
baboon	0.0001246	0.0001007	0.0001474	0.0001175	6.2078237901 × 10^−5^
barbara	0.0001317	0.0001752	0.0001240	9.5289199533 × 10^−5^	7.0011508332 × 10^−5^
cameraman	0.0001796	0.0001611	0.0001006	9.8649176392e × 10^−5^	8.0869347378 × 10^−5^
lena	0.0001380	0.0001681	0.0001196	8.9136782626e × 10^−5^	6.3427345646 × 10^−5^
peppers	0.0001242	0.0001971	0.0001118	0.0001234	7.4385365087 × 10^−5^
plane	0.0001819	0.0001447	0.0001532	7.4707859548 × 10^−5^	5.1055016436 × 10^−5^
saturn	0.0001673	0.0001524	0.0002356	0.0001255	0.0001545
woman	0.0001307	0.0001239	0.0001501	9.6613189674 × 10^−5^	6.2780501489 × 10^−5^

**Table 6 biomimetics-08-00418-t006:** Average execution time for JADE, SHADE, TRDE, SAF-DE, BDE and HSDE.

Images	K	Execution	Times (s)					
		JADE	SHADE	TRDE	SAF-DE	BDE	HSDE	Rank
baboon	2	1.44564	1.46456	1.41345	1.67615	1.97935	1.45685	1
	3	1.46766	1.48766	1.44564	1.84343	1.64564	1.44565	1
	4	1.68786	1.44646	1.44866	1.64886	1.98676	2.24568	2
	5	1.58675	1.44545	1.44564	1.77846	2.78678	2.86762	2
	6	1.53436	1.45346	1.45334	1.45444	2.13454	2.34535	1
barbara	2	1.45341	1.43482	1.42345	1.74332	1.78734	1.45434	1
	3	1.57863	1.48312	1.44563	1.85536	1.97883	1.44532	2
	4	1.51456	1.41651	1.41616	1.85687	2.04564	2.54554	1
	5	1.51536	1.45465	1.45443	1.95434	2.05446	2.34535	1
	6	1.54533	1.44535	1.42463	1.94564	2.14534	2.34565	1
cameraman	2	1.48434	1.44535	1.44533	1.85433	1.85435	1.53155	1
	3	1.55643	1.45645	1.54564	1.94564	1.94334	1.55334	2
	4	1.54534	1.45465	1.45646	1.95645	2.35645	2.05645	2
	5	1.76466	1.45666	1.58466	2.18466	2.15648	2.50546	2
	6	1.66465	1.54566	157646	2.14564	2.24645	2.54564	2
lena	2	1.54646	1.55646	1.45645	1.85643	1.94566	1.54561	1
	3	1.56448	1.54646	1.44665	1.74878	1.45648	1.45651	1
	4	1.56468	1.56465	1.66466	1.95654	2.44564	2.44569	3
	5	1.61235	1.54564	1.64566	2.3464	2.14564	2.46466	3
	6	1.68645	1.53455	1.56464	2.45646	2.24345	2.45345	2
peppers	2	1.95646	1.96456	1.84654	2.35465	2.86460	1.95640	1
	3	1.46456	1.96465	1.86464	2.64564	2.36465	1.95643	2
	4	2.05464	1.95465	1.45646	2.64565	2.64455	2.45662	1
	5	2.31534	2.54344	1.94533	2.55345	2.41546	2.85606	1
	6	2.53434	2.34533	2.13545	2.75345	2.45344	2.45647	1
plane	2	1.64646	1.45456	1.74538	1.87546	2.05445	1.55466	4
	3	1.84645	1.54556	1.64560	1.91454	1.94564	1.54565	3
	4	1.61556	1.54646	1.66456	1.95645	2.64665	2.53455	3
	5	1.73453	1.54534	1.45343	2.04344	2.25434	2.35453	1
	6	1.65546	1.54567	1.54564	2.06464	2.24567	2.44564	1
saturn	2	1.68645	1.54654	1.56488	1.98648	1.93543	1.55433	2
	3	1.74545	1.68463	1.63543	2.13535	2.15343	1.83543	1
	4	1.74564	1.64556	1.64564	2.16456	2.35645	2.55466	2
	5	2.14564	1.75465	1.75463	2.16546	2.34566	2.64864	1
	6	1.95465	1.84645	1.75646	2.25646	2.45643	2.75456	1
woman	2	1.95646	1.94543	1.85645	2.24564	2.46455	2.04645	1
	3	2.54564	2.35645	2.35642	2.66465	2.25646	2.65646	1
	4	2.98468	1.61556	1.54564	1.75648	2.46457	2.06456	1
	5	1.68466	1.56464	1.58468	2.18678	2.17867	2.48442	2
	6	1.78498	1.64848	1.54846	2.18468	2.58468	2.24564	1

**Table 7 biomimetics-08-00418-t007:** Standard deviation of the fitness value (k = 2, 3, 4, 5, 6).

Images	K		Standard Deviation of Fitness Value					
		JADE	SHADE	TRDE	SAF-DE	BDE	HSDE	Rank
baboon	2	0	0.03543	0	0.012354	0	0.04564	1
	3	0.01453	0.03453	0.00434	0.00564	0.00478	0.04698	1
	4	0.04552	0.04678	0.00135	0.04345	0.04544	0.01045	1
	5	0	0.04538	0.00456	0.00879	0.06453	0.03455	2
	6	0.01345	0.04344	0.00345	0.00456	0.03455	0.00134	2
barbara	2	0.03464	0.14563	0	0.04538	0.05464	0.08687	1
	3	0.05464	0.34575	0.04564	0.05456	0.35644	0.15312	1
	4	0.04568	0.34565	0.03456	0.04564	0.24564	0.07875	1
	5	0.05468	0.37545	0.03436	0.04565	0.45344	0.07645	1
	6	0.04567	0.56542	0.01345	0.08785	0.04634	0.03785	1
cameraman	2	0.05464	0.04564	0.00154	0.03456	0.01456	8.61 × 10^−6^	2
	3	0.00785	0.04564	0.00325	0.00453	0.04534	0.00354	1
	4	0.56546	0.45687	0.57544	0.64843	0.68454	0.69434	3
	5	0.01587	0.03455	0.00534	0.00045	0.01543	0.02543	2
	6	0.00854	0.02345	0.00454	0.03446	0.04566	0.00456	1
lena	2	0.05464	0.34564	0.00546	0.54644	0.14545	0.05944	1
	3	0.05466	0.45664	0.02365	0.14665	0.37869	0.66780	1
	4	0.15646	0.24564	0.02654	0.06544	0.53565	0.06656	1
	5	0.07456	0.14646	0.06545	0.04545	0.15765	0.05687	4
	6	0.04383	0.11346	0.02456	0.00767	0.04564	0.01546	2
peppers	2	0.01646	0.34377	0.01343	0.24668	0.05445	0.14546	1
	3	0.04566	0.15464	0.00645	0.14566	0.14566	0.06456	1
	4	0.05430	0.16464	0.03434	0.06458	0.14645	0.04555	1
	5	0.04455	0.17354	0.01544	0.35434	0.04556	0.05462	1
	6	0.04645	0.08783	0.02345	0.07376	0.55388	0.03453	1
plane	2	0.04345	0.34564	0.00876	0.27866	0.38686	0.06778	1
	3	0.03324	0.23453	0.00454	0.00876	0.34535	0.15434	1
	4	0.08554	0.15464	0.03545	0.04535	0.12455	0.15344	2
	5	0.04445	0.25435	0.04561	0.03456	0.25464	0.0345	3
	6	0.04565	0.45345	0.06771	0.06452	0.04562	0.0545	5
saturn	2	0.03245	0.04545	0	0.04355	0	0.03454	1
	3	0.04564	0.05464	0.04554	0.04565	0.54434	0.04543	2
	4	0.09754	0.14564	0.04544	0.15465	0.04564	0.05445	1
	5	0.14455	0.09787	0.05644	0.04564	0.24564	0.15444	2
	6	0.02453	0.24534	0.00454	0.05645	0.14534	0.07456	1
woman	2	0.15654	0.15767	0.00112	0.15642	0.00456	0.14567	1
	3	0.05464	0.15644	0.05244	0.35644	0.34564	0.00025	2
	4	0.05456	0.45644	0.04567	0.25465	0.05464	0.23457	1
	5	0.05784	0.15644	0.01544	0.00042	0.00645	0.05434	2
	6	0.04544	0.15464	0.04244	0.15464	0.01544	0.09798	2

## Data Availability

Data openly available in a public repository. The data that support the findings of this study are openly available in Berkeley Split Dataset at https://www2.eecs.berkeley.edu/Research/Projects/CS/vision/bsds/ (accessed on 20 June 2023).

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
