# Peer review of "Otsu Multi-Threshold Image Segmentation Based on Adaptive Double-Mutation Differential Evolution"

_biomimetics, 2023, doi:10.3390/biomimetics8050418_

Round 1
Reviewer 1 Report
I have the following concern.
1. The use of differential evolution in genetic algorithms is well-known. Specify your difference or feature of application.
2. In addition to the Otcu method, there are other image segmentation algorithms, such as Prewitt, Watershed, Sobel, Canny, Robert, Log, and others. This should be mentioned in the introduction.
3. It is necessary to give quantitative estimates of the comparison of your approach with other image segmentation methods.
4. It is known that in addition to SSIM, MSE, PSRN, other similarity metrics are used to evaluate the similarity of images, for example MS-SSIM, UOI, VIF, ERGAS and others. You need to pay attention to this and justify why you did not use them.
5. The second time segmentation results of different methods given in appendices A do not make sense.
6.References must be supplemented with articles for 2022-2023 to confirm the relevance of your approach to image segmentation.
Minor editing of English language required
Author Response
REVIEWER 1:
Reviewer 1: In response to the first reviewer's comments, we thank him/her for his/her helpful comments. We have made the following revisions in response to these comments.
- Comment
The use of differential evolution in genetic algorithms is well-known. Specify your difference or feature of application.
Responses:
Agreed and Modified. (See Introduction. We have added in the introduction the DE and its differential performance in different application fields.)
Differential Evolution Algorithm (DE) is a population intelligent optimization algorithm that searches for the optimal solution by simulating the process of natural evolution. With the continuous deepening of research on differential evolution algorithms, their application fields are gradually expanding. More and more optimization problems are combined with differential evolution algorithms to solve the optimal solution, such as scheduling problems, traveling salesman problems and image processing.In scheduling problems, differential evolution algorithms optimize scheduling strategies and resource allocation to maximize work efficiency or minimize work time. Differential evolution combined with scheduling problems can handle complex constraint problems and quickly converge to find the optimal solution. The traveling salesman problem is a classic combinatorial optimization problem that requires finding the shortest path so that the traveling salesman can visit several cities and ultimately return to the starting city. The differential evolution algorithm searches for the optimal solution by adjusting the arrangement order of cities, with the goal of minimizing travel and finding the optimal solution for the shortest path. In this problem, differential evolution algorithm can efficiently search for the global optimal solution. In image processing problems, differential evolution algorithms are used for image registration, image segmentation, feature extraction, and image enhancement.
Particle swarm optimization simulates the collective behavior of bird swarms. In image segmentation, each particle represents a possible segmentation scheme, and through continuous iteration, seeks the best segmentation result. Ant colony algorithm simulates the behavior of ants when searching for food, applies it to image segmentation, and gradually obtains the best segmentation results through pheromone propagation and reinforcement strategies. Genetic algorithm simulates the evolution process in nature, iterating through selection, crossover, and mutation operations to continuously iterate the population, maintaining its diversity and exploration ability through mutation, and ultimately finding the optimal solution. The differential evolution algorithm exhibits good performance and advantages in image segmentation due to its limited control parameters, strong robustness, and difficulty in falling into optimal solutions.
- Comment
In addition to the Otsu method, there are other image segmentation algorithms, such as Prewitt, Watershed, Sobel, Canny, Robert, Log, and others. This should be mentioned in the introduction.
Responses:
Agreed and Modified. (See Introduction. We have already elaborated on the image segmentation methods of Prewitt, Sobel ,Watershed, Robert, and Canny in the Introduction.)
In the revised manuscript, We introduce five image segmentation methods(Prewitt, Sobel,Watershed, Robert,Canny), The Prewitt image segmentation method is a classic edge detection algorithm used to extract edge information from images. This method is based on gradient calculation of the grayscale values of image pixels, and determines the edges in the image by detecting gradient changes. Watershed image segmentation method is a region based image segmentation method. In this method, the image is considered as a terrain map, where the height represents the grayscale value of the image. Image segmentation is achieved by finding watersheds, which divide the image into different regions or sets of different regions. The advantage of this method is that it can preserve areas with clear edges and rich textures in the image; However, there is also a drawback of over segmentation for images with large uniform areas. Canny image segmentation method is a classic edge detection algorithm that can accurately detect edges in an image. This algorithm determines image edges by detecting gradient changes, and improves the accuracy and continuity of edges through non maximum suppression, dual threshold processing, and edge connectivity. Sobel image segmentation method is a commonly used edge detection algorithm that is used to detect edges in images. The Sobel operator is used to calculate the gradient value of image pixels to determine the position and direction of the image. The Robert image segmentation method is a classic edge detection algorithm that applies the Robert operator to convolution the image and calculates the gradient values between adjacent pixels to determine the position and direction of the edges. Log image segmentation method is an image segmentation method based on Laplacian Gaussian filter. This method detects edges and other structures in an image by performing Gaussian and Laplacian filtering operations on the image
- Comment
It is necessary to give quantitative estimates of the comparison of your approach with other image segmentation methods.
Responses:
Agreed and Modified. (See 4. In the revised manuscript, We added experimental results of the TRDE algorithm and four other image segmentation algorithms of Prewitt, Sobel, Robert, Canny under four different image quality evaluation indicators.)
In the revised manuscript, Table 3 shows the Structural Similarity Index (SSIM) results of the TRDE algorithm applied to image segmentation compared to other image segmentation methods.Table 4 shows the structural similarity index (MS-SSIM) results of the TRDE algorithm applied to image segmentation compared to other image segmentation methods. Compared with traditional SSIM, MS-SSIM compares the similarity of images at multiple scales and weights the similarity at different scales to obtain the final evaluation result. The improved differential evolution algorithm in this article is used for threshold image segmentation, selecting an appropriate threshold to preserve the structural information of the original image as much as possible after segmentation. Due to the possible presence of noise or rough edges in the images segmented by edge detection algorithms, the structural similarity between the segmentation results and the original image may decrease. Therefore, the experimental results of the SSIM and MS-SSIM indicators for the segmented images of the TRDE algorithm in Tables 3 and 4 are higher than those of the other four edge detection methods (Prewitt, Sobel, Canny, Robert).
Table 5 shows the similarity index (UQI) of the TRDE algorithm applied to image segmentation compared to other image segmentation methods. The UQI value range is between 0 and 1, and a higher value indicates that the two images are more similar and the image quality is better. The UQI indicator comprehensively considers image mean, variance, and covariance, which can effectively evaluate the similarity and quality between images. The experimental results of the image quality evaluation index UQI are shown in Table 5. From the table, we can see that compared with other image segmentation methods, the TRDE algorithm ranks high in most cases. For example, in images such as Barbara and Lena, the Canny algorithm ranks first and the TRDE algorithm ranks second. In cameraman images, TRDE algorithm ranks first. Overall, TRDE has strong stability when applied to image segmentation algorithms, resulting in good segmentation image quality.
- Comment
It is known that in addition to SSIM, MSE, PSNR, other similarity metrics are used to evaluate the similarity of images, for example MS-SSIM, UOI, VIF, ERGAS and others. You need to pay attention to this and justify why you did not use them.
Responses:
Agreed and Modified. (See 4. We added similarity quality evaluation indicators (SSIM, MS-SSIM, UQI) results between the TRDE algorithm and other image segmentation methods.)
Thank you for your valuable feedback. In the revised manuscript, Table 3 shows the Structural Similarity Index (SSIM) results of the TRDE algorithm applied to image segmentation compared to other image segmentation methods.Table 4 shows the structural similarity index (MS-SSIM) results of the TRDE algorithm applied to image segmentation compared to other image segmentation methods. Compared with traditional SSIM, MS-SSIM compares the similarity of images at multiple scales and weights the similarity at different scales to obtain the final evaluation result.
Table 5 shows the similarity index (UQI) of the TRDE algorithm applied to image segmentation compared to other image segmentation methods. The UQI value range is between 0 and 1, and a higher value indicates that the two images are more similar and the image quality is better. The UQI indicator comprehensively considers image mean, variance, and covariance, which can effectively evaluate the similarity and quality between images. The experimental results of the image quality evaluation index UQI are shown in Table 5. From the table, we can see that compared with other image segmentation methods, the TRDE algorithm ranks high in most cases. For example, in images such as Barbara and Lena, the Canny algorithm ranks first and the TRDE algorithm ranks second. In cameraman images, TRDE algorithm ranks first. Overall, TRDE has strong stability when applied to image segmentation algorithms, resulting in good segmentation image quality.
ERGAS (Relative Global Error in Synthesis) is a quality metric used to evaluate image synthesis or reconstruction, typically used to compare the differences between synthesized and original images. It takes into account the differences in overall structure and information. However, threshold segmentation is a simple segmentation method based on pixel grayscale values, without involving complex image synthesis or reconstruction processes. Threshold segmentation divides image pixels into two parts and only classifies them based on their grayscale values, without considering the structure and contextual information of the image.
Therefore, as an indicator for evaluating image synthesis or reconstruction, the ERGAS indicator is not suitable for evaluating binary images after threshold segmentation.
- Comment
The second time segmentation results of different methods given in appendices A do not make sense.
Responses:
Agreed and Modified. (In the revised manuscript, we have clarified the meanings of Figures 1, 2, 3, and 4.)
Thank you for your valuable feedback. In the revised manuscript, Figure 1 shows the image segmentation results obtained by the TRDE algorithm at 5 thresholds within the range of 8 images [2,6]. Figure 2 shows the image segmentation results obtained by the BDE algorithm at 5 thresholds within the range of 8 images [2,6]. Figure 3 shows the image segmentation results obtained by the JADE algorithm at 5 thresholds within the range of 8 images [2,6]. Figure 4 shows the image segmentation results obtained by the SAF-DE algorithm at 5 thresholds within the range of 8 images [2,6].
- Comment
References must be supplemented with articles for 2022-2023 to confirm the relevance of your approach to image segmentation.
Responses:
Agreed and Modified. (See reference. Update the list with some 2022-2023 research in the revised manuscript)
We have updated the references and added references from 2022-2023 onwards to the literature.
- Abualigah L, Almotairi K H, Elaziz M A. Multilevel thresholding image segmentation using meta-heuristic optimization algorithms: Comparative analysis, open challenges and new trends[J]. Applied Intelligence, 2023, 53(10): 11654-11704.
- Su H, Zhao D, Elmannai H, et al. Multilevel threshold image segmentation for COVID-19 chest radiography: A framework using horizontal and vertical multiverse optimization[J]. Computers in Biology and Medicine, 2022, 146: 105618.
- Han Y, Chen W, Heidari A A, et al. Multi-verse optimizer with rosenbrock and diffusion mechanisms for multilevel threshold image segmentation from COVID-19 chest X-Ray images[J]. Journal of bionic engineering, 2023, 20(3): 1198-1262.
- Mahajan S, Mittal N, Salgotra R, et al. An efficient adaptive salp swarm algorithm using type II fuzzy entropy for multilevel thresholding image segmentation[J]. Computational and Mathematical Methods in Medicine, 2022, 2022.
- Zhou, Yuan, et al. "An improved firefly algorithm for task scheduling in cloud computing environment." IEEE Access 10 (2022): 2363-2372.
- Yang, Xin-She, and Suash Deb. "Cuckoo search via Lévy flights." Nature-inspired algorithms for optimisation. Springer, Berlin, Heidelberg, 2022. 123-144.
- Sun Y, Yang Y. An Adaptive Bi-Mutation-Based Differential Evolution Algorithm for Multi-Threshold Image Segmentation[J]. Applied Sciences, 2022, 12(11): 5759.
- Guo, Yalong, et al. "Image segmentation based on a hybrid artificial bee colony algorithm with SSIM and PSNR evaluation." Engineering with Computers (2022): 1-16.
- Sun, Yihua, et al. "Differential evolution algorithm with SSIM and PSNR metrics for image segmentation." Multimedia Tools and Applications 81.10 (2022): 15577-15594.
- Comment
Minor editing of English language required
Responses:
We tried our best to improve the manuscript and made some changes to the manuscript. These changes will not influence the content and framework of the paper. And here we did not list the changes but marked in red in the revised paper. We appreciate for Editors/Reviewers' warm work earnestly and hope that the correction will meet with approval.
Reviewer 2 Report
The article deals with the problem of image segmentation as a process of dividing an image into several non-overlapping regions and selecting target regions of interest from them.
One of the well-known methods of image segmentation is threshold segmentation, which is quite simple and effective. Threshold segmentation can be either single-threshold or multi-threshold. In 1978, Nobuyuki Otsu proposed a threshold image segmentation method which was published in January 1979. This method is based on the idea of separating image pixels into background and target pixels based on the halftone characteristics of the image. This separation is based on solving the problem of maximizing the variance between the target and background images. This means that the threshold image segmentation method can be treated as an optimization problem. The problem here is that when solving such a problem, the time complexity increases exponentially with the number of thresholds and the amount of computation increases. As a possible way to solve this problem, the paper proposes to use differential evolution with double mutation based on adaptive control parameters, as well as a mechanism for finding adaptive control parameters and a double mutation strategy.
Experimental results show that this approach has good robustness. In addition, this algorithm has higher image segmentation accuracy and lower time complexity compared to other algorithms for solving the segmentation problem.
The obtained results are of interest for specialists in the field of computer vision and image processing. However, the article in the presented form is not without shortcomings. In particular, one can observe some carelessness of its presentation. Here are some examples illustrating this statement.
The article is devoted to the development of an idea proposed by Nobuyuki Otsu in January 1979 (there is a typographical error in the text of the article, the year 1997 is indicated for this publication). However, the bibliographic description of Nobuyuki Otsu's article in the reference list is incomplete and inaccurate:
18. Ostu N , Nobuyuki O , Otsu N .A thresholding selection method from gray level histogram[J]. 1979.
The full description of this article is as follows:
Otsu, N. A thresholding selection method from gray-level histograms. IEEE Trans. on Systems, Man, and Cybernetics. 1979, 9, 62-66.
It is also worth noting the careless design of the reference list. In addition to the above mentioned incomplete and inaccurate description of Nobuyuki Otsu's article, similar incompleteness is observed in items 9, 16, 18, 19, and 24. The rest of the entries do not meet the format for presenting bibliographic descriptions in MDPI journals.
There are also carelessness and misprints in the article, such as incorrect and misspelled surnames of the authors of the publications:
For example, in the text of the article (line 47) we see the following:
Differential evolution [12] (DE) was proposed by storm and price in 1997.
It should be as follows:
Differential evolution [12] (DE) was proposed by Storn and Price in 1997.
In this regard, the authors should be advised to carefully read the text of the article and eliminate the existing inaccuracies in it.
Another remark concerns the presentation of formulas in the text. In different formulas, symbols differ in size, and this size does not always correlate with the size of symbols in the main text. As a result, the text looks somewhat sloppy, visually imperfect. It would be better, of course, to use a LaTeX presentation, where such a situation cannot exist by definition, or to somehow adjust the existing typesetting.
In general, the article is of certain interest for those who are related to image processing and can be published after elimination of the noted shortcomings.
Author Response
REVIEWER 2:
Reviewer 2: In response to the third reviewer's comments, we thank him/her for his/her helpful comments. We have made the following revisions in response to these comments.
Comment
The article deals with the problem of image segmentation as a process of dividing an image into several non-overlapping regions and selecting target regions of interest from them.
One of the well-known methods of image segmentation is threshold segmentation, which is quite simple and effective. Threshold segmentation can be either single-threshold or multi-threshold. In 1978, Nobuyuki Otsu proposed a threshold image segmentation method which was published in January 1979. This method is based on the idea of separating image pixels into background and target pixels based on the halftone characteristics of the image. This separation is based on solving the problem of maximizing the variance between the target and background images. This means that the threshold image segmentation method can be treated as an optimization problem. The problem here is that when solving such a problem, the time complexity increases exponentially with the number of thresholds and the amount of computation increases. As a possible way to solve this problem, the paper proposes to use differential evolution with double mutation based on adaptive control parameters, as well as a mechanism for finding adaptive control parameters and a double mutation strategy.
Experimental results show that this approach has good robustness. In addition, this algorithm has higher image segmentation accuracy and lower time complexity compared to other algorithms for solving the segmentation problem.
The obtained results are of interest for specialists in the field of computer vision and image processing. However, the article in the presented form is not without shortcomings. In particular, one can observe some carelessness of its presentation. Here are some examples illustrating this statement.
The article is devoted to the development of an idea proposed by Nobuyuki Otsu in January 1979 (there is a typographical error in the text of the article, the year 1997 is indicated for this publication). However, the bibliographic description of Nobuyuki Otsu's article in the reference list is incomplete and inaccurate:
Ostu N , Nobuyuki O , Otsu N .A thresholding selection method from gray level histogram[J]. 1979.
The full description of this article is as follows:
Otsu, N. A thresholding selection method from gray-level histograms. IEEE Trans. on Systems, Man, and Cybernetics. 1979, 9, 62-66.
It is also worth noting the careless design of the reference list. In addition to the above mentioned incomplete and inaccurate description of Nobuyuki Otsu's article, similar incompleteness is observed in items 9, 16, 18, 19, and 24. The rest of the entries do not meet the format for presenting bibliographic descriptions in MDPI journals.
There are also carelessness and misprints in the article, such as incorrect and misspelled surnames of the authors of the publications:
For example, in the text of the article (line 47) we see the following:
Differential evolution [12] (DE) was proposed by storm and price in 1997.
It should be as follows:
Differential evolution [12] (DE) was proposed by Storn and Price in 1997.
In this regard, the authors should be advised to carefully read the text of the article and eliminate the existing inaccuracies in it.
Another remark concerns the presentation of formulas in the text. In different formulas, symbols differ in size, and this size does not always correlate with the size of symbols in the main text. As a result, the text looks somewhat sloppy, visually imperfect. It would be better, of course, to use a LaTeX presentation, where such a situation cannot exist by definition, or to somehow adjust the existing typesetting.
In general, the article is of certain interest for those who are related to image processing and can be published after elimination of the noted shortcomings.
Responses:
Agreed and Modified. (In the revised manuscript, we have corrected formatting errors such as references and formulas.)
Thank you for your careful inspection. We apologize for our carelessness. Based on your feedback, We recheck all the formulas in the article for any errors and reorder them to show them all to the left. Correct formatting errors in the references in the article. Check the article for incorrect words and correct them.
Round 2
Reviewer 1 Report
I am almost satisfied with the answers to my concerns. The changes and additions made have significantly improved the perception of the obtained results.
Minor editing of English language required